# Electroacupuncture versus sham electroacupuncture in treating low anterior resection syndrome after rectal cancer surgery: Study protocol for a randomized controlled trial

Zi-Yue Wang[1], Guang-Xia Shi[1], Yu Wang[1], Wei Pei[2], Ying-Chi Yang[3], Jing-Wen Yang[1], Xiao-Ya Wei[1], Jin-Ying Jia[4]*, Jian-Feng Tu[1]*, Cun-Zhi Liu[1]

1 International Acupuncture and Moxibustion Innovation Institute, School of Acupuncture-Moxibustion and Tuina, Beijing University of Chinese Medicine, Beijing, China, 2 Department of Colorectal Surgery, National Cancer Center/National Clinical Research Center for Cancer/Cancer Hospital, Chinese Academy of Medical Sciences and Peking Union Medical College, Beijing, China, 3 Beijing Friendship Hospital, Capital Medical University, Beijing, China, 4 Department of Traditional Chinese medicine, The First Affiliated Hospital of Zhengzhou University, Zhengzhou, China

* tujianfeng1@126.com (JFT); AA13525552922@163.com (JYJ)

## Abstract

### Purpose

Low anterior resection syndrome (LARS), a frequent postoperative complication of rectal resection, has been shown to significantly impact patients' quality of life. Electroacupuncture (EA), a non-pharmaceutical treatment, ameliorates gastrointestinal symptoms and promotes bowel movement. However, high-quality clinical evidence is lacking. This study aimed to determine whether EA can improve LARS symptoms compared with sham electroacupuncture (SA).

### Study design and methods

This multicenter, randomized, sham-controlled clinical trial will be carried out across the outpatient clinics of three tertiary medical centers in China. A total of 136 patients with LARS who meet the inclusion criteria will be randomly allocated, in equal proportions (1:1), to either the EA or SA group. Each patient will undergo treatment three times weekly during the first four weeks and twice-weekly sessions over the subsequent four weeks. After the intervention, a 24-week follow-up period will be conducted. The primary outcome is the change in the LARS score from baseline to the end of week 8. Secondary outcomes include changes in the LARS score at other time points, response rate of patients showing reduced defecation dysfunction, subjective distress related to intestinal symptoms, Wexner Diarrhea score, Bristol Bowel Diary, EORTC-QLQ-C30 Quality of Life Questionnaire, and Fecal Incontinence Quality of Life Scale.

**Data availability statement:** No datasets were generated or analyzed in the course of this study. Upon completion of the study, all pertinent data will be made accessible.

**Funding:** This research was funded by the Qihuang Scholar Project, which is part of the National Administration of Traditional Chinese Medicine (China), accessible at http://www.natcm.gov.cn/. The funding organization had no involvement in the study design, data collection, analysis, publication decisions, or manuscript preparation. Liu CZ is the recipient of this funding. Additionally, no grant numbers are associated with this funding.

**Competing interests:** All authors affirm that they have no conflicts of interest associated with the publication of this study.

## Discussion

This study will provide evidence of EA from multiple perspectives, investigate its potential application in LARS after rectal cancer surgery, and guide the development of therapy tailored to meet specific individual health needs.

## Ethics and dissemination

Ethical approval for this study was granted by the Ethics Committee of Beijing University of Chinese Medicine (No. 2024BZYLL0402). All participants enrolled in the trial will provide written informed consent prior to randomization. Results will be prepared for submission to a peer-reviewed academic journal.

## Trial registration

ITMCTR2024000195. International Traditional Medicine Clinical Trial Registry (http://itmctr.ccebtcm.org.cn/zh-CN/Home/ProjectView?pid=badb9af5-248b-4818-b3b6-25002f0fa0d5).

## Introduction

Low anterior resection (LAR) is a common sphincter-preserving surgical procedure for lower rectal cancer, with the aim of maintaining anal sphincter function [1]. As a frequent postoperative complication of rectal surgery, low anterior resection syndrome (LARS) is characterized by significant alterations in bowel function that occur after surgery and is considered an inevitable result of rectal resection [2]. Approximately 60%−90% of rectal cancer patients develop LARS symptoms following surgical intervention [2–5]. LARS is characterized by a range of bowel functions, including fecal incontinence, a sense of urgency, and other defecatory disturbances [6]. While advancements in surgical procedures and neoadjuvant treatment strategies have significantly improved patient survival rates, the incidence of rectal cancer continues to increase, with an increasing trend among younger individuals [7,8]. These factors undoubtedly contribute to substantial psychological and physical distress in patients' daily lives [7–11].

Currently, various treatments are recommended for LARS, including dietary modifications and sacral nerve stimulation. Among these, dietary intervention is widely recognized as the primary therapeutic strategy for patients with LARS [10]. However, long-term adherence to dietary modifications requires that patients maintain good dietary habits, necessitating strong self-discipline, which can be challenging for many patients. Given the similarity in symptoms between LARS and the diarrhea-predominant subtype of irritable bowel syndrome (IBS-D), medications used for IBS-D are often used in the treatment of LARS. In cases of urgent defecation, loperamide [11] and 5-hydroxytryptamine receptor antagonists [12] are administered to enhance fluid absorption in the intestine and alleviate defecatory urgency. Although these medications demonstrate some efficacy in managing cancer-associated diarrhea, they

must be used with caution [13]. Stimulation of the sacral nerve is an effective treatment for LARS [14]. However, it is not the first-line intervention because of its invasive nature, high cost, and risk of complications [15]. Enema, recognized for its simplicity and low-cost treatment for LARS, is particularly suitable for symptoms related to colonic motility disorders, such as frequent defecation [16]. Nevertheless, potential side effects, such as rectal bleeding and perianal pain, may occur [17]. Pelvic floor muscle training (PFMT) [3] is a low-cost minimally invasive intervention that can improve defecatory symptoms. However, its clinical application remains limited because of the lack of standardized training protocols and low patient compliance. Therefore, safer and more effective therapies are urgently required to relieve these symptoms.

As a complementary and alternative treatment, acupuncture has garnered increased attention because of its potential role in improving postoperative gastrointestinal function [18]. Previous studies have demonstrated that EA has a positive impact on the postoperative recovery of gastrointestinal function in patients undergoing rectal cancer surgery [19–22]. Acupuncture demonstrates clear advantages in alleviating postoperative ileus, reducing abdominal distension and pain, and significantly improving patients' quality of life [23–25]. Although some studies suggest that acupuncture may be an effective intervention for managing defecation dysfunction following low rectal cancer resection, the current evidence remains insufficient to draw definitive conclusions [26]. Consequently, the clinical application of acupuncture for LARS remains uncertain and requires further research to clarify its therapeutic effects. To bridge this research gap, we are conducting a clinical trial aimed at assessing whether EA can improve symptoms in patients with LARS compared to sham SA.

## Materials and methods

### Study design

This multicenter, randomized controlled trial was scheduled to be conducted at three tertiary hospitals (Beijing Friendship Hospital Affiliated to Capital Medical University, Cancer Hospital of Chinese Academy of Medical Sciences, and the First Affiliated Hospital of Zhengzhou University). A total of 136 eligible patients with LARS will be randomly assigned in a 1:1 ratio to either the EA group (n = 68) or the SA group (n = 68). The trial has received ethical approval and has been registered (ITMCTR2024000195) with the International Traditional Medicine Clinical Trial Registration Platform. The study protocol (V.2.0 February 2024) will be documented following the Standard Protocol Items: Recommendations for Interventional Trials (SPIRIT) [27]. A summary of the SPIRIT schedule is presented in Fig 1. S1 File contains the completed SPIRIT checklist, while S2 File includes the original, ethics-approved version of the study protocol.

### Trial status

This clinical trial is presently undergoing participant recruitment. The first participant was enrolled on August 22, 2024, following trial registration, with the enrollment period expected to be completed by December 2025. If participant recruitment is not completed within the planned timeframe, we will consider extending the recruitment period or increasing the number of centers as appropriate.

### Participants

Patients scheduled to undergo rectal resection for rectal cancer will be approached and provided with detailed information about the trial prior to surgery. Those who express an interest in participating will be considered potential candidates, and their baseline information will be documented. The Clinical Research Coordinator (CRC) will conduct postsurgical interviews with potential participants to evaluate eligibility based on inclusion and exclusion criteria. Participants deemed eligible will be asked to voluntarily sign a written informed consent (provided in the S3 File) prior to random assignment.

#### Inclusion criteria

1. Adults aged 18–75 years, regardless of gender.

2. The diagnosis of LARS was confirmed at least one month after rectal resection or stoma closure (LARS score ≥ 21).

| | STUDY PERIOD | | | | | | | | |
|---|---|---|---|---|---|---|---|---|---|
| | Enrolment | Allocation | Post-allocation | | | | | | Close-out |
| TIMEPOINT | Week -1 | Week 0 | AFT | Week 2 | Week 4 | Week 6 | Week 8 | Week 12 | Week 24 |
| **ENROLMENT:** | | | | | | | | | |
| Informed consent | × | | | | | | | | |
| Eligibility screening | × | | | | | | | | |
| Preparation period | ←———————→ | | | | | | | | |
| Allocation | | × | | | | | | | |
| **INTERVENTIONS:** | | | | | | | | | |
| EA | | ←——————————————————→ | | | | | | | |
| SA | | ←——————————————————→ | | | | | | | |
| **ASSESSMENT:** | | | | | | | | | |
| LARS score* | | | | | | | × | | |
| LARS score | × | | | × | × | × | × | × | × |
| Proportion of patients with reduced degree of bowel dysfunction | | × | | × | × | × | × | × | × |
| Subjective distress of bowel symptoms | | × | | | × | | × | × | × |
| EORTC-QLQ-C30 | | × | | | × | | × | × | × |
| Bristol Bowel Scale | | × | | | × | | × | × | × |
| FIQL | | × | | | × | | × | × | × |
| Wexner score for diarrhea | | × | | | × | | × | × | × |
| Credibility/expectation evaluation | | × | | | | | | | |
| Blinding | | | × | | × | | | | |
| Adverse events | | ←—————————————————————————————————→ | | | | | | | |
| Rescue medicine | | ←—————————————————————————————————→ | | | | | | | |

**Fig 1. SPIRIT schedule.** *, Primary outcome. AFT, after the first treatment; FIQL, Fecal Incontinence Quality of Life Scale; EORTC-QLQ-C30, EORTC-QLQ-C30Quality of Life Questionnaire; EA, electroacupuncture; SA, sham electroacupuncture.

3. Sign a written informed consent voluntarily.

**Exclusion criteria**

1. History of other types of colorectal cancer surgery or resection of other intestinal segments, including Hartmann's procedure, abdominopelvic resection, transanal endoscopic microsurgical excision, or surgical removal of the sigmoid colon.

2. History of other pelvic surgeries for non-neoplastic conditions.

3. Preoperative fecal incontinence.

4. History of psychiatric illness, chronic alcohol dependence, or substance misuse.

5. Diagnosis of inflammatory bowel disease or irritable bowel syndrome.

6. Received acupuncture treatment within the past one months.

7. Use of implantable electronic medical devices, including but not limited to cardiac pacemakers.

8. Participation in another clinical study.

## Allocation and blinding

A stratified block randomization method will be utilized, with study centers serving as the stratification factor. Eligible patients will be randomly assigned at a 1:1 ratio to either the EA group or the SA group. The randomization sequences will be generated using SAS software by statisticians who will not be involved in outcome assessment. These randomization sequences will be securely maintained by a randomization administrator who will not participate in the trial intervention, assessment, or data analysis. Upon the inclusion of an eligible patient, the acupuncturists will obtain a randomization assignment by phone from the randomization administrator. To maintain allocation concealment, the randomization sequence will not be accessible to any other members during the trial period. While the acupuncturists will not be blinded, blinding will be maintained for patients, outcome assessors, and statisticians. After the first treatment session, we will assess the blinding, as well as participants' expectations and perceived credibility of the intervention.

## Trial withdrawal

Patients who withdraw from the study after randomization for any reason and no longer participate in the intervention regimen or observation period will be considered dropouts. Upon dropout, the investigator is expected to make all reasonable attempts to reach the individual, document the cause of withdrawal, and complete relevant assessments. All data related to the dropout cases should be preserved appropriately for documentation.

## Interventions

**EA group.** Licensed acupuncturists with a minimum of three years of clinical practice will administer the EA intervention. Before the trial begins, all acupuncturists involved in the study will complete a standardized training program to ensure intervention consistency. Eligible patients will undergo a total of 20 EA sessions over eight weeks, with 30 minutes per session. The intervention protocol will adhere to Standards for Reporting Interventions in Clinical Trials of Acupuncture (STRICTA) guidelines [28].

Based on a previous study, EA will be administered at the following bilateral treatment acupoints: Zhongliao (BL33), Ciliao (BL32), Sanyinjiao (SP6), Taixi (KI3), Tianshu (ST25), Shuidao (ST28), Zusanli (ST36), and Yinlingquan (SP9) [29]. The locations of these acupoints are determined in accordance with the National standard GB/T 12346−2021 "Name and Location of Meridians and Acupoints". Detailed descriptions of the eight specific acupoints are presented in Table 1 and Fig 2.

Image reprinted from the [3Dbody application] with authorization form [3Dbody (Shanghai) Digital Technology Co., Ltd.], under a Creative Commons Attribution (CC BY) license. Copyright [2024]. The picture is authorized by 3Dbody application.

Both the acupoints and the acupuncturist's hands will be cleansed using 75% alcohol. Sterile, disposable Hwato-brand acupuncture needles (0.30 mm × 40 mm or 0.25 mm × 30 mm) will be used. Needles will be manually manipulated for 30 seconds to achieve the sensation of "deqi" (a feeling of numbness, soreness, distension, or pain induced by acupuncture). A portable electroacupuncture apparatus (Hwato, SDZ-V) will be used in this trial. In the first session,

**Table 1. Location of acupoints in EA group.**

| Acupoints | Locations |
|---|---|
| Zhongliao (BL33) | In the sacral region, opposite the third sacral posterior foramen |
| Ciliao (BL32) | In the sacral region, the second sacral posterior foramen |
| Sanyinjiao (SP6) | On the medial of the lower leg, 3 cun[a] above the tip of the medial malleolus, posterior to the medial margin of the tibia |
| Taixi (KI3) | On the medial side of the foot, in the depression between the medial tip of the ankle and the Achilles tendon |
| Tianshu (ST25) | On the abdomen, level with the umbilicus, 2 cun lateral to the anterior midline |
| Shuidao (ST28) | On the abdomen, 3 cun below the umbilicus, 2 cun lateral to the anterior midline |
| Zusanli (ST36) | On the anterolateral side of the lower leg, 3 cun below the Dubi (ST35) point, one finger-width (middle finger) lateral to the anterior crest of the tibia |
| Yinlingquan (SP9) | On the medial side of the lower leg, in the depression between the inferior border of the medial condyle of the tibia and the medial border of the tibia |

[a]1 cun (≈20 mm) is defined as the width of the interphalangeal joint of patient's thumb.

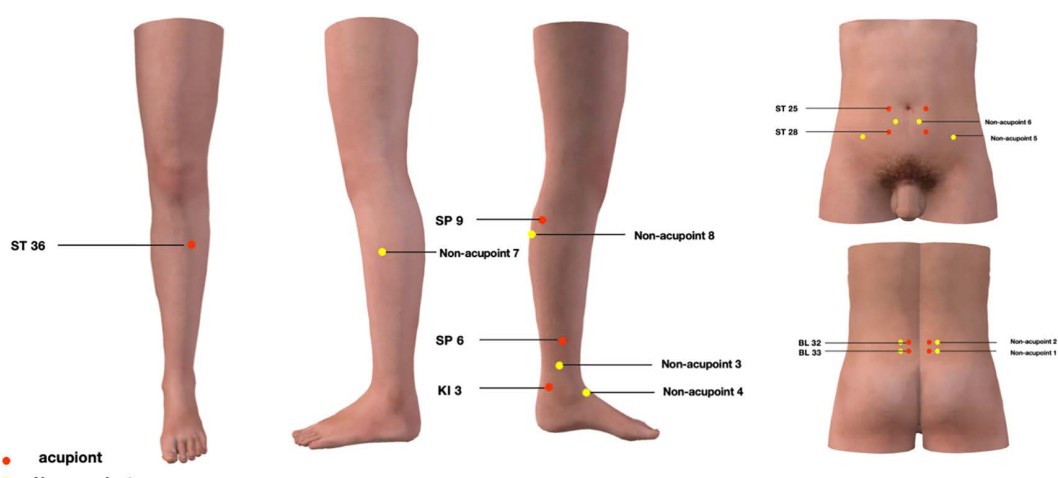

**Fig 2. Locations of acupoints and non-acupoints.**

paired electrodes will be connected to the needle handles at bilateral BL32, SP6 and KI3. In the second session, the electrodes will be applied to bilateral ST25 and ST36. These two sessions will be alternated. The current intensity will be adjusted to the maximum level at which a slight tremor appears at the needle handle, using a continuous wave frequency set at 10 Hz [30]. Needles will remain in place for a total of 30 minutes. Further details regarding EA are provided in Table 2.

**SA group.** Superficial needling will be performed at non-acupoints without any manual manipulation or "deqi" sensation. The eight non-acupoint locations are shown in Fig 2 and Table 3. Non-acupoints 1–4 and non-acupoints 5–8 will be alternated for treatment. Sham paired electrodes will be connected to non-acupoints 4 and non-acupoints 8 bilaterally, with no current applied. All the other settings will be the same as in the EA group. Further details regarding SA are provided in Table 2.

**Table 2. Details of the electroacupuncture and sham electroacupuncture intervention.**

| Item | Electroacupuncture | Sham electroacupuncture |
|---|---|---|
| Needling instrument | length: 30−40 mm, diameter: 0.25–0.3 mm; Hwato, Suzhou, China | length: 30−40 mm, diameter: 0.25–0.3 mm; Hwato, Suzhou, China |
| Retaining time | 30min | 30min |
| Treatment sessions | 20 | 20 |
| Frequency | three times a week for the first four weeks and twice a week for the last four weeks | three times a week for the first four weeks and twice a week for the last four weeks |
| Manipulation | After the needle inserted, lift, thrust and thrill smoothly to achieve De qi | – |
| Electrical acupoints | Wire 1: bilateral BL32. Wire 2, 3: bilateral SP6 and KI3. | Wire 1: bilateral non-acupoints 4. Wire 2: bilateral non-acupoints 8. |
| Electric parameter | Continuous wave; 10 Hz; The intensity of the current is set to the limit where a slight tremor appears at the needle handle; (Hwato, SDZ-V). | – |

**Table 3. Location of acupoints in SA group.**

| Acupoints | Locations |
|---|---|
| non-acupoint 1 | In the sacral region, 1 cun[a] lateral to the Zhongliao (BL33) point |
| non-acupoint 2 | In the sacral region, 1 cun lateral to the Ciliao (BL32) point |
| non-acupoint 3 | On the lower limb, 2 cun above the medial malleolus, the medial side of the tibia is median, between the Liver and Spleen meridians |
| non-acupoint 4 | On the lower limb, at the midpoint of the line connecting Qiuxu (GB40) and Jiexi (ST41) points |
| non-acupoint 5 | On the abdomen, 2 cun below the umbilicus and 1 cun apart from the midline |
| non-acupoint 6 | On the abdomen, 2 cun above the anterior superior iliac spine |
| non-acupoint 7 | On the lower limb, on the lateral side of the lower leg, 3 cun below the Yanglingquan (GB34) point, between the Gallbladder and Bladder meridians |
| non-acupoint 8 | On the lower limb, on the medial side of the lower leg, 1 cun lateral to the Chengjin (BL56) point, between the Bladder and Stomach meridians |

[a]1 cun (≈20 mm) is defined as the width of the interphalangeal joint of patient's thumb.

## Outcomes

### Primary outcome

The primary outcome of this study is the change in LARS score from baseline at week 8. The LARS score [31] is a validated and widely adopted tool for evaluating symptom severity and its effect on patients' quality of life. It comprises 5 questions addressing the following key symptoms: incontinence of flatus, leakage of liquid stool, urgency, increased stool frequency, and clustering (defined as multiple bowel movements occurring within a short timeframe). Based on the total score, patients are classified into three categories: no LARS (0–20), minor LARS (21–29), and major LARS (30–42). Due to its ease of use and strong reliability, the LARS score has become a standard assessment tool in both clinical and

research settings. The tool is particularly valuable for the postoperative follow-up of patients with rectal cancer, aiding in long-term management and improving outcomes related to bowel function.

### Secondary outcomes

1. Changes in LARS Score from Baseline at Week 2, 4, 6, 12 and 24.

   Changes in the LARS score from baseline will be assessed at weeks 2, 4, 6, 12, and 24.

2.  Proportion of Patients with Improvement in Bowel Dysfunction

At weeks 2, 4, 6, 8, 12, and 24, the number of patients improving from severe LARS to mild LARS, mild to no LARS, and severe to no LARS will be determined based on LARS score.

3. Subjective distress of intestinal symptoms

We will assess the subjective distress of intestinal symptoms using a Numerical Rating Scale (NRS) at weeks 4, 8, 12, and 24, where 0 indicates no distress, and 10 represents the worst imaginable distress.

4. EORTC-QLQ-C30 [32,33] quality of life questionnaire

The European Organization for Research and Treatment of Cancer Core Quality of Life Questionnaire (EORTC-QLQ-C30) is a validated instrument consisting of 30 items [34]. It includes 5 functional scales, three symptoms' subscales, one global health status scale, and six single-item measures [35]. Higher scores on the functional scales reflect improved quality of life, whereas elevated scores on the symptom scales represent greater symptom burden and reduce quality of life. The patients will be evaluated at weeks 4, 8, 12, and 24.

5. Bristol stool scale

At weeks 0, 4, 8, 12 and 24, we will record the frequency of defecation, stool form, and urgency over the past week.

6. Fecal Incontinence Quality of Life Scale (FIQL)

We will evaluate the quality of life of patients with fecal incontinence, covering 4 aspects: lifestyle changes, coping/behavioral limitations, depression/self-perception, and social embarrassment at weeks 0, 4, 8, 12, and 24.

7. Wexner diarrhea score

The Wexner Score, also referred to as the Cleveland Clinic Incontinence Score, is a widely utilized tool for quantifying the severity of fecal incontinence. It evaluates five parameters: the frequency of incontinence to gas, leakage of liquid and solid stool, need for pads, and lifestyle alterations. Each category is scored from 0 (no issues) to 4 (severe symptoms), with a maximum cumulative score of 20. A higher total score indicates more severe incontinence. The Wexner Score is commonly applied in clinical settings to inform treatment decisions and monitor patient progress. The evaluation will be performed at 0, 4, 8, 12, and 24 weeks.

   Rescue medicine: Patients with severe fecal incontinence will be administered Loperamide Hydrochloride Capsules (Imodium, Xi'an Janssen Pharmaceutical Co., LTD). The duration and dosage of Imodium will be recorded.

### Adverse events

During acupuncture, bleeding, subcutaneous hematoma, numbness, soreness, or distension after needle insertion, as well as other adverse reactions, will be recorded in the case report form (CRF). All adverse events (AEs) will be addressed with appropriate symptomatic treatment. If the acupuncturist is unable to resolve the issue, consultation with an appropriate medical specialist will be required for further management.

 

## Data availability and management statement

Data entry personnel and outcome assessors will receive standardized training in data management procedures. All participants' information will be documented in the original CRFs. Two individuals will promptly input the data into Excel spreadsheets and verify the accuracy of the entries against each other. In cases where data are not properly recorded, data managers will make corrections as required. All study-related documents in paper form will be securely archived, while digital files will be stored on password-protected computers to ensure data security. All study materials, in any format, will be retained for a minimum of five years following publication. If readers have inquiries about the published findings, they may reach out to the corresponding author to request access to the original dataset. Participants' personal details will be kept confidential.

## Quality control

We will implement standardized training for researchers across all centers to ensure consistency and adherence to study protocols. The training curriculum will include a comprehensive introduction to the study's objectives and requirements, mastery of the relevant diagnostic and treatment standards, randomization procedures, acupuncture techniques, and the proper use of evaluation forms. Training programs will be tailored to researchers based on their specific roles, ensuring that individuals performing the same role across different research centers successfully complete a consistent assessment before proceeding with the trial. Furthermore, we will establish a research center management team headed by the principal investigator, with center inspectors responsible for oversight and quality control. Rigorous documentation and reporting protocols will be enforced throughout the clinical trial. Each center will utilize uniformly printed CRFs, which will be uniquely numbered. The study operations strictly adhere to the designed protocol, and CRFs will be completed carefully and objectively, accurately documenting any issues encountered during the clinical trial.

## Sample size

According to findings from the same study, the anticipated reduction in LARS score from baseline is 6.8 points in the EA group and 4.1 points in the SA group, assuming a standard deviation of 5 and a two-tailed α level of 0.05. Analysis determined that 54 participants per group are required to detect a statistically significant difference. Accounting for a 20% dropout rate, the target enrollment has been increased to 68 participants per group. As a result, the total planned sample size for this study is 136 participants.

## Statistical analysis

All statistical analyses will be performed using SPSS software version 20.0, with $P < 0.05$ considered indicative of statistical significance. Continuous data will be summarized either as mean ± standard deviation (M ± SD) or as median with interquartile range. Categorical variables will be presented as either frequencies or percentages.

This study will utilize a modified intention-to-treat (mITT) analysis, which includes every participant who has undergone at least one intervention after randomization. The primary outcome will be analysed using a two-sample t-test (α = 0.05). If normality cannot be assumed, the Mann–Whitney $U$ test will be employed. To assess the robustness of the primary outcome, a sensitivity analysis was performed using a linear regression model, with baseline LARS score, age, sex, stoma status, and study center included as covariates [36,37]. To address the missing data, multiple imputation will be applied. Mixed-effects models will be employed for between-group comparisons of all repeated measures of continuous outcomes, with baseline values included as covariates. For categorical variables, chi-square ($\chi^2$) tests will be conducted. A sensitivity analysis will be performed for the primary outcome. In this study, the PP set is defined as participants who have received at least 16 treatments without any apparent protocol violations.

## Patient and public involvement

Patients and members of the public are not involved in the design, recruitment, or conduct of this clinical trial. Upon study completion, the results will be shared with all patients.

## Discussion

The findings from this trial aim to offer high-quality evidence for the potential benefits of EA for LARS following rectal cancer surgery. The study will be carried out in full compliance with the protocol. Standard patient care for rectal cancer will be maintained throughout the study to ensure that our findings can be generalized to most hospitals worldwide. Puncture will be administered to all participants, which is expected to enhance their treatment adherence. To evaluate treatment responses, this study will incorporate rigorous scientific methods along with comprehensive outcome measures. Additionally, these methodological approaches will ensure the accurate reporting of EA's efficacy in managing LARS upon the completion of the trial.

Previous studies have confirmed the high prevalence of LARS. Defecation dysfunction has a significant impact on patients' daily lives and overall well-being, leading to substantial physical and psychological burdens [38,39]. For the 20%−40% of patients who experience persistent symptoms, long-term pharmacological management of bowel dysfunction remains challenging and often proves insufficient for achieving optimal symptom control [2]. More rigorous research is essential to optimize therapies and alleviate LARS-related morbidities. Nonpharmacological interventions have gained attention as essential treatment strategies for managing LARS. Evidence suggests that acupuncture enhances parasympathetic function while reducing sympathetic hyperactivity [40–42]. By modulating the enteric nervous system through the vagal and sympathetic pathways, acupuncture helps restore autonomic balance, which is often disrupted following surgery [43]. In addition, acupuncture directly stimulates the sacral nerve roots, reduces visceral hypersensitivity, improves pelvic floor muscle function, and regulates sensory thresholds in the rectum [14]. These findings provide valuable insights into the role of acupuncture as a therapeutic approach for LARS.

One limitation of this study is that blinding of the acupuncturists is not feasible. LARS symptoms exhibit significant individual variability, with differences in the patients' physiological conditions, surgical techniques, and postoperative rehabilitation influencing their response to EA treatment. The lack of adequate consideration of individual variability may compromise the external validity of the study. Furthermore, the trial does not clearly delineate whether EA exerts synergistic or independent effects when combined with other therapies, which may limit the accurate assessment of its true efficacy.

## Supporting information

**S1 File. SPIRIT checklist.**
(DOC)

**S2 File. Original protocol approved by the ethics committee.**
(DOCX)

**S3 File. Model consent form.**
(DOCX)

## Acknowledgments

We gratefully acknowledge the contributions of all trial participants and recruitment site staff involved in this study.

## Author contributions

**Conceptualization:** Cun-Zhi Liu.

**Data curation:** Cun-Zhi Liu.

**Funding acquisition:** Guang-Xia Shi, Jian-Feng Tu.

**Investigation:** Zi-Yue Wang, Yu Wang.

**Methodology:** Zi-Yue Wang, Guang-Xia Shi, Yu Wang, Jian-Feng Tu.

**Project administration:** Cun-Zhi Liu.

**Resources:** Cun-Zhi Liu.

**Supervision:** Jian-Feng Tu.

**Validation:** Guang-Xia Shi, Jian-Feng Tu.

**Writing – original draft:** Zi-Yue Wang.

**Writing – review & editing:** Guang-Xia Shi, Yu Wang, Wei Pei, Ying-Chi Yang, Jing-Wen Yang, Xiao-Ya Wei, Jin-Ying Jia, Jian-Feng Tu, Cun-Zhi Liu.

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
