## [Decision Letter · Decision Letter 0]

Dear Dr. Tu,

Thank you for submitting your manuscript to PLOS ONE. After careful consideration, we feel that it has merit but does not fully meet PLOS ONE’s publication criteria as it currently stands. Therefore, we invite you to submit a revised version of the manuscript that addresses the points raised during the review process.

Thank you for submitting the following manuscript to PLOS ONE.

Please revise the manuscript according to the reviewers' comments and upload the revised file.

We look forward to receiving your revised manuscript.

Kind regards,

Yung-Hsiang Chen, Ph.D.

Academic Editor

PLOS ONE

Journal Requirements:

*2.* When completing the data availability statement of the submission form, you indicated that you will make your data available on acceptance. We strongly recommend all authors decide on a data sharing plan before acceptance, as the process can be lengthy and hold up publication timelines. Please note that, though access restrictions are acceptable now, your entire data will need to be made freely accessible if your manuscript is accepted for publication. This policy applies to all data except where public deposition would breach compliance with the protocol approved by your research ethics board. If you are unable to adhere to our open data policy, please kindly revise your statement to explain your reasoning and we will seek the editor's input on an exemption. Please be assured that, once you have provided your new statement, the assessment of your exemption will not hold up the peer review process.

Additional Editor Comments:

Thank you for submitting the following manuscript to PLOS ONE.

Please revise the manuscript according to the reviewers' comments and upload the revised file.

Reviewers' comments:

Reviewer's Responses to Questions

**Comments to the Author**

1. Does the manuscript provide a valid rationale for the proposed study, with clearly identified and justified research questions?

Reviewer #1: Yes

Reviewer #2: Yes

2. Is the protocol technically sound and planned in a manner that will lead to a meaningful outcome and allow testing the stated hypotheses?

Reviewer #1: Yes

Reviewer #2: Yes

3. Is the methodology feasible and described in sufficient detail to allow the work to be replicable?

Reviewer #1: Yes

Reviewer #2: Yes

4. Have the authors described where all data underlying the findings will be made available when the study is complete?

Reviewer #1: Yes

Reviewer #2: Yes

5. Is the manuscript presented in an intelligible fashion and written in standard English?

Reviewer #1: No

Reviewer #2: Yes

You may also provide optional suggestions and comments to authors that they might find helpful in planning their study.

Reviewer #1: This is a well-designed acupuncture study. Randomization is clearly stated as are the primary and secondary outcomes. There are some problems with the sample size computation: no clinically meaningful difference is stated, the test statistic to be used is not defined, and assumptions on the variability given. In particular, the effect size is based on a prior study rather than what would be viewed as clinically significant. There is no discussion of the assumptions required for the sample size computation. The primary outcome appears to be a LARS score, which may or may not be normally distributed, and the assumptions and form of the test statistic is not given. There is also no contingency plans given for lack of recruitment. A stratified randomization typically requires a stratified analysis.

The protocol needs a good proofreading. There are many misspellings and grammatical errors (e.g., "monition" is not a word).

Reviewer #2: The authors carried out a multicenter randomized clinical trial to examine the efficacy of electroacupuncture (EA) in managing low anterior resection syndrome (LARS) following rectal cancer surgery. However, the following issues need to be clarified.

1.Abstract: The discussion section references personalized treatment. Since this program follows a standardized acupuncture prescription, it is recommended that this terminology, personalized, should be removed for accuracy.

2.Introduction: Regarding the statement "Sacral nerve stimulation is an effective treatment for LARS," I recommend citing recent high-level evidence (e.g., a systematic review or RCT and so on) to substantiate this claim.

3.Introduction: The efficacy and limitations of sacral nerve stimulation have been mentioned in the second paragraph. The subsequent reiteration in the third paragraph appears redundant and should be omitted to improve conciseness.

4.Intervention: The 30-minute EA session duration warrants further justification. Given the postoperative status of participants, could the authors comment on: Patient tolerance for prone positioning during back acupoint stimulation.

5.Intervention: To enhance clarity and facilitate reader comprehension, it is advisable to merge Tables 2 and 4 into a single consolidated table, allowing for more direct comparison of the electroacupuncture and sham electroacupuncture.

6.Please briefly describe how to train acupuncturists in each center so that treatment is as consistent as possible among the three centers.

**Do you want your identity to be public for this peer review?** For information about this choice, including consent withdrawal, please see our Privacy Policy

Reviewer #1: No

Reviewer #2: No

---

## [Author Response · Author response to Decision Letter 1]

25 Jun 2025

Response to the academic editors

1. Response: Thank you for your detailed instructions regarding the revision and resubmission process. No changes have been made for our financial disclosure statement. We have carefully reviewed the resubmission guidelines and have updated the figure files.

Response: Thank you for your guidance. We have reviewed and revised the manuscript to ensure full compliance with PLOS ONE’s formatting and style guidelines, including the appropriate file names.

Response: Thank you for your reminder. This manuscript is a study protocol and does not involve any research data at this stage. We have added “Data availability statement” as below: “Reviewers or readers are welcome to contact the corresponding author (Jian-feng Tu) to request access to the original data.” (Line 410-412, Page 22)

Response: Thank you for your valuable guidance. Captions for all Supporting Information files have been added to the end of the manuscript and corresponding in-text citations have been updated to ensure consistency with PLOS ONE’s guidelines.

Response to the comments of Reviewer #1

1. no clinically meaningful difference is stated, the test statistic to be used is not defined, and assumptions on the variability given. In particular, the effect size is based on a prior study rather than what would be viewed as clinically significant. There is no discussion of the assumptions required for the sample size computation.

Response: Thank you for your valuable comments. We acknowledge the significance of the minimal clinically important difference (MCID) for LARS score. However, to date, no validated MCID has been established for this scale. Thus, our sample size calculation was based on previous studies. A between-group difference of 2.7 points and a standard deviation of 5 were assumed. The expected effect size (Cohen’s d) in our study is 0.54, which represents a moderate effect and is widely accepted as a clinically important difference [1].

1. Kraemer HC, Kupfer DJ. Size of treatment effects and their importance to clinical research and practice. Biol Psychiatry. 2006;59(11):990-6. Epub 20051220. doi: 10.1016/j.biopsych.2005.09.014. PubMed PMID: 16368078.

2. The primary outcome appears to be a LARS score, which may or may not be normally distributed, and the assumptions and form of the test statistic is not given.

Response: Thanks for your constructive suggestions. We defined the primary outcome in the original manuscript as the LARS score at week 8 (Line 226-227, Page 14). In accordance with previous high-quality research [1, 2], a normal distribution of the LARS score was assumed for sample size estimation in this trial. Given that the trial is currently in progress, it remains possible that the LARS score does not follow a normal distribution. Therefore, we have included appropriate statistical methods to address this possibility. Revisions to the statistical analysis are presented below:

“The primary outcome will be analysed using a two-sample t-test (α=0.05). If normality cannot be assumed, the Mann–Whitney U test will be employed.” (Lines 333-335, Page 19)

1. Meurette G, Faucheron JL, Cotte E, Denost Q, Portier G, Loriau J, et al. Low anterior resection syndrome after rectal resection management: multicentre randomized clinical trial of transanal irrigation with a dedicated device (cone catheter) versus conservative bowel management. Br J Surg. 2023;110(9):1092-5. doi: 10.1093/bjs/znad078. PubMed PMID: 36977128; PubMed Central PMCID: PMCPMC10416684.

2. Marinello FG, Jiménez LM, Talavera E, Fraccalvieri D, Alberti P, Ostiz F, et al. Percutaneous tibial nerve stimulation in patients with severe low anterior resection syndrome: randomized clinical trial. Br J Surg. 2021;108(4):380-7. doi: 10.1093/bjs/znaa171. PubMed PMID: 33793754.

3. There is also no contingency plans given for lack of recruitment.

Response: Thank you for highlighting this. We have revised the manuscript accordingly and incorporated the following information:

“If participant recruitment is not completed within the planned timeframe, we will consider extending the recruitment period or increasing the number of centers as appropriate.” (Lines 122-124, Page 7)

4. A stratified randomization typically requires a stratified analysis.

Response: Thank you for your valuable comments. We agree that stratification factors may potentially influence the results. Therefore, we have added a sensitivity analysis in which stratification variable (center) would be incorporated into the linear regression model. The manuscript has been revised as follows:

“To assess the robustness of the primary outcome, a sensitivity analysis was performed using a linear regression model, with baseline LARS score, age, sex, stoma status, and study center included as covariates [1, 2].” (Lines 335-337, Page 19)

1. Battersby NJ, Bouliotis G, Emmertsen KJ, Juul T, Glynne-Jones R, Branagan G, et al. Development and external validation of a nomogram and online tool to predict bowel dysfunction following restorative rectal cancer resection: the POLARS score. Gut. 2018;67(4):688-96. Epub 20170123. doi: 10.1136/gutjnl-2016-312695. PubMed PMID: 28115491.

2. Andersson J, Angenete E, Gellerstedt M, Angerås U, Jess P, Rosenberg J, et al. Health-related quality of life after laparoscopic and open surgery for rectal cancer in a randomized trial. Br J Surg. 2013;100(7):941-9. doi: 10.1002/bjs.9144. PubMed PMID: 23640671; PubMed Central PMCID: PMCPMC3672685.

5. The protocol needs a good proofreading. There are many misspellings and grammatical errors (e.g., "monition" is not a word).

Response: We sincerely apologize for the oversight. The revised manuscript has been thoroughly edited by a native English speaker to correct all grammatical and typographical errors. For example, to ensure more accurate and standardized terminology, the term “safety monitoring” has been corrected to “adverse events”; the spelling error “trail” has been corrected to “trial”; and the duplicated word “receive” has been removed from the Allocation and Blinding section.

Response to the comments of Reviewer #2

1. Abstract: The discussion section references personalized treatment. Since this program follows a standardized acupuncture prescription, it is recommended that this terminology, personalized, should be removed for accuracy.

Response: We apologize for this. We would like to clarify that our study was conducted based on a standardized research protocol. We made the deletion modification in the relevant position. (Line 41, Page 2)

2. Introduction: Regarding the statement "Sacral nerve stimulation is an effective treatment for LARS," I recommend citing recent high-level evidence (e.g., a systematic review or RCT and so on) to substantiate this claim.

Response: Thank you for your recommendations. We have cited the following articles in the revised manuscript. (Line 76, Page 5)

1. Huang Y, Koh CE. Sacral nerve stimulation for bowel dysfunction following low anterior resection: a systematic review and meta-analysis. Colorectal Dis. 2019;21(11):1240-8. Epub 20190613. doi: 10.1111/codi.14690. PubMed PMID: 31081580.

3. Introduction: The efficacy and limitations of sacral nerve stimulation have been mentioned in the second paragraph. The subsequent reiteration in the third paragraph appears redundant and should be omitted to improve conciseness.

Response: Thank you for your constructive suggestions. We have removed the description of sacral nerve stimulation previously included in the third paragraph of the manuscript.

4. Intervention: The 30-minute EA session duration warrants further justification. Given the postoperative status of participants, could the authors comment on: Patient tolerance for prone positioning during back acupoint stimulation.

Response: Thank for your comments. A 30-minute treatment duration is widely used in clinical acupuncture practice and in postoperative studies [1, 2]. Based on our experience, participants were enrolled one month after surgery in this trial, by which time most had recovered sufficiently to tolerate both acupuncture and prone positioning for dorsal acupoint stimulation.

1. Wang Y, Yang JW, Yan SY, Lu Y, Han JG, Pei W, et al. Electroacupuncture vs Sham Electroacupuncture in the Treatment of Postoperative Ileus After Laparoscopic Surgery for Colorectal Cancer: A Multicenter, Randomized Clinical Trial. JAMA Surg. 2023;158(1):20-7. doi: 10.1001/jamasurg.2022.5674. PubMed PMID: 36322060; PubMed Central PMCID: PMCPMC9631228.

2. Ng SSM, Leung WW, Mak TWC, Hon SSF, Li JCM, Wong CYN, et al. Electroacupuncture reduces duration of postoperative ileus after laparoscopic surgery for colorectal cancer. Gastroenterology. 2013;144(2):307-13.e1. Epub 20121106. doi: 10.1053/j.gastro.2012.10.050. PubMed PMID: 23142625.

5. Intervention: To enhance clarity and facilitate reader comprehension, it is advisable to merge Tables 2 and 4 into a single consolidated table, allowing for more direct comparison of the electroacupuncture and sham electroacupuncture.

Response: Many thanks for your comments. We combined Tables 2 and 4 into Table 2 in the manuscript.

6. Please briefly describe how to train acupuncturists in each center so that treatment is as consistent as possible among the three centers.

Response: Thank you for your valuable suggestions. To ensure consistency in the acupuncture intervention, we established standardized operating procedures and provided training for all acupuncturists. Their performance was regularly monitored, and any deviations from the protocol were addressed through retraining and correction to maintain adherence.

7. PLOS authors have the option to publish the peer review history of their article (what does this mean?). If published, this will include your full peer review and any attached files.

Response: Thank you for the information regarding the option to publish the peer review history. We acknowledge and support PLOS’s commitment to transparency in scientific publishing. As stated in the Ethics and Dissemination section of our manuscript, we are committed to open scientific practices; therefore, we have no objection to the public disclosure of the peer review history.

---

## [Decision Letter · Decision Letter 1]

Electroacupuncture versus sham electroacupuncture in treating low anterior resection syndrome after rectal cancer surgery: study protocol for a randomized controlled trial

PONE-D-25-14598R1

Dear Dr. Tu,

We’re pleased to inform you that your manuscript has been judged scientifically suitable for publication and will be formally accepted for publication once it meets all outstanding technical requirements.

Kind regards,

Yung-Hsiang Chen, Ph.D.

Academic Editor

PLOS ONE

Additional Editor Comments (optional):

Congratulations on the acceptance of your manuscript, and thank you for your interest in submitting your work to PLOS ONE.

Reviewers' comments:

Reviewer's Responses to Questions

**Comments to the Author**

1. Does the manuscript provide a valid rationale for the proposed study, with clearly identified and justified research questions?

Reviewer #1: Yes

2. Is the protocol technically sound and planned in a manner that will lead to a meaningful outcome and allow testing the stated hypotheses?

Reviewer #1: Yes

3. Is the methodology feasible and described in sufficient detail to allow the work to be replicable?

Reviewer #1: Yes

4. Have the authors described where all data underlying the findings will be made available when the study is complete?

Reviewer #1: Yes

5. Is the manuscript presented in an intelligible fashion and written in standard English?

Reviewer #1: Yes

You may also provide optional suggestions and comments to authors that they might find helpful in planning their study.

Reviewer #1: All comments have been addressed. xxxxxxxxxxxxxxxxxxxxxxxxxxxxxxxxxxxxxxxxxxxxxxxxxxxxxxxxxxxxxxxxx

**Do you want your identity to be public for this peer review?** For information about this choice, including consent withdrawal, please see our Privacy Policy

Reviewer #1: No

---

## [Editor Report · Acceptance letter]

PONE-D-25-14598R1

PLOS ONE

Dear Dr. Tu,

I'm pleased to inform you that your manuscript has been deemed suitable for publication in PLOS ONE. Congratulations! Your manuscript is now being handed over to our production team.

Kind regards,

on behalf of

Dr. Yung-Hsiang Chen

Academic Editor

PLOS ONE